# Effect of Laser Ablation on Microwave Attenuation Properties of Diamond Films

**DOI:** 10.3390/ma12223700

**Published:** 2019-11-09

**Authors:** Minghui Ding, Yanqing Liu, Xinru Lu, Weizhong Tang

**Affiliations:** 1Institute for Advanced Materials and Technology, University of Science and Technology Beijing, Beijing 100083, China; 2Institute of Interdisciplinary Information Sciences, Tsinghua University, Beijing 100084, China; liuyanqing870425@163.com (Y.L.); 18911871715@126.com (X.L.)

**Keywords:** diamond films, laser ablation, permittivity, microwave attenuation, high thermal conductivity

## Abstract

Thermal conductivity is required for developing high-power microwave technology. Diamond has the highest thermal conductivity in nature. In this study, a diamond film was synthesized by microwave plasma chemical deposition, and then long and short conductive graphite fibers were introduced to the diamond films by laser ablation. The permittivity of the samples in the K-band was measured using the transmission/reflection method. The permittivity of diamond films with short graphite fibers increased. The increase in real part of permittivity can be attributed to electron polarization, and the increase in the imaginary part can be ascribed to both polarization and electrical conductivity. The diamond films with long graphite fibers exhibited a highly pronounced anisotropy for microwave. The calculation of microwave absorption shows that reflection loss values exceeding −10 dB can be obtained in the frequency range of 21.3–23.5 GHz when the graphite fiber length is 0.7 mm and the sample thickness is 2.5 mm. Therefore, diamond films can be developed into a microwave attenuation material with extremely high thermal conductivity.

## 1. Introduction

Microwave attenuation materials are widely used in modern vacuum electronic devices. With rapid advances in this field, materials with a high thermal conductivity are needed. Traditionally, BeO- and AlN-based ceramic composites are microwave absorptive materials because of their high thermal conductivities (BeO 370 W m^−1^ K^−1^ [1] and AlN 320 W m^−1^ K^−1^ [2]). The lossy fillers of these materials can be SiC [3], carbon fibers [4,5,6,7,8], and so on. Because of the interface thermal resistance between the lossy fillers and ceramic matrix, which drastically increases the phonon scattering, thermal conductivity of these composites decreases [8]. For example, BeO-SiC composite is a material with a thermal conductivity of ~160 W m^−1^ K^−1^ [9].

The thermal conductivity of diamond (~2000 W cm^−1^ K^−1^) is far higher than that of BeO and AlN. Therefore, diamond is practically used for several high-temperature applications [10]. For example, diamond–copper composites with a thermal conductivity of 226–742 W cm^−1^ K^−1^ are fabricated by sintering diamond and copper particles for heat spreader materials and packages [11]. The thermal conductivity of diamond-based composites has been reported in the literature [12]. However, no diamond-based composites are reported as microwave absorptive materials, probably because of the high electrical conductivity of these materials [13]. 

Rather than a matrix, types of diamond particles can be used as lossy fillers. For instance, n-diamond particles with an electrical conductivity close to the metallic character prepared from catalyzed carbon black were used in a microwave absorption material [14,15]. In the literature [16], nano-diamond particles with a mean diameter of 3–10 nm were also used as a filler component. However, in composites where diamond materials are used as fillers, its high thermal conductivity makes no sense because of the small amount.

Except for the granular diamond materials used for sintering, diamond films have been reported as a bulk diamond material produced by microwave plasma chemical vapor deposition (MPCVD). However, in general, the loss tangent of pure diamond films is only in the range of 10^−3^–10^−5^ [17,18,19,20]. That means pure diamond films are a microwave transparent material, usually used in high-power microwave windows [21].

To make diamond films microwave absorptive, lossy filler components should be added. In our previous study [19,20,22], nitrogen- and boron-doped diamond films were prepared by MPCVD. It was found that the loss tangent of nitrogen-doped diamond films was close to the pure diamond films, i.e., after nitrogen doping, the diamond films are also a microwave transparent material [19,20]. This result was consistent with the previous study [17]. 

On the other hand, diamond films can be transformed from a microwave transparent material into a microwave lossy material when boron atoms are doped into the diamond films. Mechanism analysis has shown that the increase in the real part of permittivity mainly resulted from the hopping polarization of bound charges, while the increase in the imaginary part resulted from both hopping polarization and valence band conduction [22]. However, as a semiconductor, the electrical conductivity of boron-doped diamond films will significantly increase with increasing temperature [23], affecting its attenuation properties at high temperatures.

As a carbon allotrope, graphite can be used as lossy fillers because of its high electrical conductivity in some ceramic matrix composites such as flake graphite [24]. In addition, when the composites contain carbon materials, the absorptive performances can be improved by the graphitization of carbon, enhancing the filler’s electrical conductivity [4].

For diamond films, graphite can be easily added using some methods such as laser ablation [25,26,27,28], high temperature [29], and oxidation [30]. Among these methods, laser ablation is flexible and cost-effective. In addition, it can be used for local accurate graphitization [28]. In general, lasers operated at a short region below the fundamental absorption band of diamond at 225 nm are preferable because of high optical absorption [31]. However, when the energy of lasers surpasses the threshold, the ablation proceeds via surface graphitization, providing strong absorption in the UV-IR range. This makes the ablation rate is wavelength-independent [27].

In this paper, graphite was introduced into free-standing diamond films using laser ablation. Microwave properties of these films were measured in the K-band (18–26.5 GHz). Diamond films can be transformed into microwave absorbing material by laser ablation. This discovery suggests that such diamond-based materials with a high thermal conductivity could find applications as microwave attenuation materials.

## 2. Materials and Methods

### 2.1. Synthesis of a Free-Standing Diamond Film by MPCVD

A diamond film was deposited on a (100) single crystal silicon substrate of 30 mm in diameter using a MPCVD reactor [32]. The silicon substrate was first scratched uniformly with 10-μm diamond powders and then ultrasonically cleaned in acetone and methanol for 10 min sequentially. As process gases, methane and hydrogen were introduced into the reactor using gas mass flow controllers. The deposition conditions and dimension of diamond film are shown in Table 1.

After deposition, the diamond film was separated from the silicon substrate by acid etching. Then, the removed film was mechanically polished from both sides mechanically and laser cut into rectangular samples of 10.6 × 4.3 mm^2^ size. Finally, the samples were acid cleaned to remove possible contaminants.

### 2.2. Preparation of Diamond Samples Using Laser Ablation

An original Nd:YAP laser system (1.06 μm wavelength, 500 μs pulse duration, 200 Hz pulse repetition rate, 20 W output power) was used in laser ablation experiments. The spatial profile of the laser beam was nearly Gaussian, and the laser radiation was focused on diamond surface into a spot of ~80 µm in size. The laser beam was scanned along a programmed path only one time. Diamond films were placed on a computer-driven X-Y stage and permitted to displace under the laser beam with a scanning velocity of 90 mm min^−1^.

Because it is possible for lasers to locally heat and ablate diamond material, diverse graphite-based graphics can be introduced into diamond materials such as a line, a flake, or even more complex graphics. For the sake of convenience, only a shape of line graphite was introduced into the diamond material in this study, referred to as graphite fiber (GF), like carbon fiber (CF). Both the GF and CF have high electrical conductivities [26,33]. Therefore, GF could be studied to some extent in terms of CF.

There are mainly two main types of CF according to the geometry, continuous CF, and short CF [33]. Continuous CF is used to improve both the properties of electromagnetism and mechanics, but it exhibits an anisotropy under microwave [5,34]. However, similar to traditional metallic frequency selective surface (FSS), continuous CFs can be paved perpendicularly crosslinked with each other, which is isotropic for measurements [34,35]. Unlike continuous CF, a short CF can be sparsely dispersed in a composite and has been extensively studied for microwave absorption [3,4,5,6,7]. However, it is difficult to form randomly distributed short GFs inside a diamond using laser ablation.

According to the previous discussions about CF, continuous and short GFs were introduced to diamond films using laser ablation. A schematic diagram of laser ablation traces of samples is shown in Figure 1.

As shown in Figure 1, GFs were periodically arranged on the surface of diamond films, except for sample A, in which no GFs existed. Samples B and C exhibited continuous GFs, but the directions of GFs of these two samples were vertical to each other. Samples D and E used short GFs, and those short GFs formed periodic grids similar to FSS.

Regarding the decision of GF’s length, it should be noted that when the microwave length is larger than the grid period, the periodic modulation of material properties can be homogenized using effective material properties [36]. The frequency wavelength of K-band was 11.3–16.7 mm, larger than the sample size (10.6 mm × 4.3 mm), and it was impossible to insert FSS structures into the samples. Thus, the microwave scattering of the periodic structures can be ignored and the frequency response of samples was only related to the complex permittivity. Therefore, according to the size of sample, the length of GF was determined as shown in Figure 1.

### 2.3. Characterization of Diamond Films

Surface structure and quality of diamond films were characterized by laser confocal microscopy (Olympus, OLS4000, Tokyo, Japan), scanning electron microscopy (SEM, LEO4500, Jena, Germany), and Raman spectroscopy. Raman spectroscopy was carried out using a Horiba HR-800 Raman spectrometer (Paris, France) with a 532-nm wavelength laser at room temperature. The in-plane thermal conductivity of samples was measured using the photothermal deflection technique, as described in detail in the literature [37]. The permittivity of the samples was measured using the transmission/reflection (T/R) technique using an Agilent N5244A vector network analyzer (Palo Alto, CA, USA) [38].

The device used for the T/R measurement is schematically shown in Figure 2a. The device measures the reflection and transmission of a microwave reference signal when it passes through the material in a waveguide. In the T/R measurement of this study, the surfaces of all samples were ablated by laser faced to the incident wave, as shown in Figure 2a. In Figure 2b, a practical measurement system is shown.

## 3. Results and Discussions

### 3.1. Morphologies and Raman Spectra of Diamond Films

The morphologies of samples B–E are shown in Figure 3a–d. As shown, surface blacking caused by laser ablation appeared, similarly to the results reported in the literature [27]. The morphology of a trace ablated by laser is shown in Figure 3e. As shown, a groove with a rough wall and constant width along the laser trace appeared.

The depth of grooves measured by laser confocal microscopy is shown in Figure 4. The depth of grooves was about 50–60 μm, and the width was about 90 μm. This indicates that during ablation, a part of carbon atoms was removed. This can be explained as follows. The wavelength of Nd:YAP laser used in this study was 1.06 µm (~1.2 eV) and it was lower than the band gap of diamond (5.4 eV). In this situation, the laser ablation proceeded via surface graphitization [27,28,31]. In other words, the graphite was generated at diamond surface in the early parts of a laser pulse, and then the energy of the latter part of laser pulse was deposited and localized in the graphite layer, leading to material evaporation [28,31]. Thus, the nearly Gaussian cross-section of GFs can be obtained by laser ablation.

It should be noted that, in this study, the cross-section of graphite layer was nearly Gaussian, determining the diameter and structure of GFs. Based on the results reported in the literature [39], the diameter and structure of short CFs would affect its microwave absorption properties, and GFs may be similar to CFs. However, in this study, the effects of diameter and structure of GFs on microwave absorption properties were only dicussed in the introduction of GF itself and effects of its length and orientation.

Figure 5 shows the Raman spectra of a diamond film obtained from the virgin region surface (curve a) and from the bottom of laser-ablated area (curve b). In curve a, the characteristic diamond peak (1332 cm^−1^) was sharp and no obvious D peak (1335 cm^−1^) or G peak (1580 cm^−1^) was observed. The 1420 cm^−1^ peak that appeared in Raman spectrum was a [N-V]^0^ related fluorescence peak. This peak was not an intrinsic Raman peak, which would disappear by increasing or decreasing the excitation wavelengths [40]. The presence of this peak indicates that a small number of nitrogen atoms were added to the chamber of MPCVD system. On one hand, the incorporation of nitrogen impurities reduce the quality of diamond films [17,19,20]. On the other hand, the impurities facilitate the absorption of laser energy when the photon energy of laser is lower than the band gap of diamond and contribute to the graphitization of diamond [31].

In curve B, the characteristic peak of diamond was absent for laser-ablated film and the spectrum was clearly dominated by the typical structure of amorphous carbon phase with G and D bands. This indicates that a new layer of graphite formed due to laser ablation [41]. This new layer was responsible for a new property, namely, very high electrical conductivity [26].

The above results indicate that GFs combined with conductive graphite were successfully introduced to diamond films using laser ablation.

In this study, the thermal conductivity of all samples was measured at room temperature. For sample A, the virgin diamond film had the highest thermal conductivity, ~1800 W cm^−1^ K^−1^, among all samples. The decrease in thermal conductivity relative to pure CVD diamond can be attributed to the scattering of phonons by nitrogen impurities [10]. The thermal conductivity of samples B–E was lower than that of sample A due to the damage of structure by laser ablation. However, the damage layer was shallow (~60 µm). Therefore, the effect was not severe. For example, the thermal conductivity of sample E was 1600 W cm^−1^ K^−1^, only about 10% reduction, i.e., after laser ablation, the diamond films also exhibited extreme thermal conductivity, which was far higher than that of ceramic matrix composites.

### 3.2. Dielectric Properties

The electromagnetic parameters of the diamond films with GFs are mainly determined by their permittivity ($\varepsilon_{r}=\varepsilon_{r}^{\prime }-j\varepsilon_{r}^{\prime \prime }$) and permeability ($\mu_{r}=\mu_{r}^{\prime }-j\mu_{r}^{\prime \prime }$). Because diamond and graphite are nonmagnetic materials, their permeability is normal, i.e., $\mu_{r}=1$. Therefore, only permittivity should be discussed. The permittivity and loss tangent (tan δ = $\varepsilon_{r}^{\prime \prime }/\varepsilon_{r}^{\prime }$) of diamond films were measured in the K-band and the results are shown in Figure 6.

In Figure 6, sample A without laser ablation had the lowest permittivity, and the permittivity of samples increased after ablation, indicating that laser ablation changed the microwave response of diamond films. The permittivity of sample C is not shown in Figure 6 because of the failure in the measurement of T/R method.

As shown in Figure 6a,b, the real part of permittivity of sample A was 5.5, which remained almost constant in the entire K-band. The value of the imaginary part of permittivity was close to zero. The results were similar with those reported in the literature [17,18,19,20]. For sample B where the GFs were vertical to the incident electric field, both the real and imaginary parts of permittivity slightly increased along with the loss tangent, as shown in Figure 6. To explain this change in permittivity, the dielectric polarization of solid materials should be considered in detail. Permittivity of a dielectric material can be explained by Debye theory, as follows [22,42]:(1)$$\varepsilon_{r}^{\prime }=\varepsilon_{\infty }+\frac{\varepsilon_{0}-\varepsilon_{\infty }}{1+\omega^{2}\tau^{2}}$$
(2)$$\varepsilon_{r}^{\prime \prime }=\frac{\sigma }{\omega \varepsilon_{0}}+\frac{\varepsilon_{0}-\varepsilon_{\infty }}{1+\omega^{2}\tau^{2}}\omega \tau$$
where ω is the microwave angular frequency, τ is the relaxation time, σ is the electrical conductivity, ε_0_ is static permittivity, and $\varepsilon_{\infty }$ is the relative dielectric permittivity at high-frequency limit.

The first term $\varepsilon_{\infty }$ in Equation (1) arose from electric polarization. It was a constant for all five samples because only carbon atoms contributed to this polarization, i.e., for sample A, $\varepsilon_{r}^{\prime }$ = $\varepsilon_{\infty }$ = 5.5. The second term in Equation (1) arose from other potential polarizations, such as atomic polarization, reorientation polarization, and space charge polarization [43]. For sample B, the slight increase in the real part of permittivity meant that the potential polarization was weak and can be ignored.

As shown in Equation (2), the imaginary part of permittivity arose from polarizations and electrical conductivity. The slight increase in the imaginary part of permittivity of sample B indicated that the contribution from electrical conductivity was also weak, even though the GFs were conductive. This was because the GFs of sample B were perpendicular to the incident electric field and these fibers are isolated from each other. It was difficult for free electrons to pass through the gap, as the electrons can only move at the interface of GFs, resulting in a low conductivity.

As shown in Figure 6a, the real parts of permittivity of samples D and E were far higher than that of sample A. According to Equation (1), new polarizations occurred. Regarding short CFs composites, this new polarization was electron relaxation polarization [7,42,44]. This was because the free electrons in GFs responded rapidly to an alternating microwave [7,44].

Figure 6a also shows that the real part of permittivity increased with the increase in the length of GFs. The total length of GFs in sample D was 1.2-times than that of sample E (ignoring the GFs that were vertical to the incident electric field due to the slight effect on polarization). This can be explained by the longer shift paths for the free electrons provided by long GFs. The longer shift paths stimulated further polarization of the sample [44].

As shown in Figure 6b, the imaginary part of sample E was higher than that of sample D. This was mainly because of the high electrical conductivity caused by the long GFs. In addition, according to Equation (2), the electron polarization also played a role in the imaginary part.

The permittivity of sample C with almost the same total length of GFs as sample B (0.96:1) could not be measured. This was because of the strong reflection in the T/R measurement.

Figure 7 shows that the value of |S_11_| was close to 1 in the entire K-band, i.e., almost all the incident microwave was reflected. This was because the GFs of sample C were parallel to the incident electric field and free electrons moved along the GFs, resulting in a large electrical conductivity. Thus, sample C exhibited strong reflection characteristics. However, according to the reports in literature [34,35], continuous CF were used as a lossy filler and successfully improved the properties of microwave absorption. The difference between these two results can be mainly attributed to the fiber spacing, as discussed in Section 2.2.

Figure 6 and Figure 7 show that the dielectric properties of diamond films with short GFs were better than those of diamond films with continuous GFs. On the other hand, the permittivity of the former can be adjusted by changing the length of GF. In addition, the results of samples B and C show that the continuous GFs exhibited anisotropy under microwave. This phenomenon was similar to the results of continuous CF composites [5,34]. Unfortunately, anisotropic dielectric materials have many limitations in practical applications.

### 3.3. Microwave Absorbing Property

The microwave reflection loss (RL) curves were calculated using the following equations [42]:(3)$$\mathrm{RL}=20\log_{10}\left(\frac{Z_{\mathrm{in}}-Z_{0}}{Z_{\mathrm{in}}+Z_{0}}\right)$$
(4)$$Z_{\mathrm{in}}=\sqrt{\frac{\mu_{r}}{\varepsilon_{r}}}\tan h\left(\left(j\frac{2\pi fL}{c}\right)\sqrt{\varepsilon_{r}\mu_{r}}\right)$$
where *f* is the frequency of microwave, *L* is the thickness of absorber, *c* is the velocity of light, *Z*_0_ is the impedance of free space; *Z*_in_ is the input impedance of material, and *ε*_r_ and *µ*_r_ are the relative permittivity and permeability, respectively. The RL value of −10 dB was equivalent to 90% efficiency of microwave absorption.

In this study, sample A had low loss and was usually used as microwave transparent windows, and sample C was strongly reflective. None of these samples were suitable for microwave attenuation. Thus, only the RL curves of samples B, D, and E are shown in Figure 8.

The RL of samples with continuous GFs (sample B) and short GFs (samples D and E) with different thicknesses was calculated as shown in Figure 8. It can be observed from Figure 8 that the samples with short GFs had better microwave absorptive performance than the sample with continuous GFs. As shown in Figure 8a, the RL of sample B was poor and the maximum RL was lower than −10 dB. This was mainly because of the low loss tangent of sample B (Figure 6c).

As shown in Figure 8b, for sample D, the RL peak shifted from 25.6 GHz to 19.1 GHz with an increase in thickness from 2.2 mm to 3 mm. Less than −10 dB was obtained for this range of thickness. This can be attributed to a quarter-wavelength attenuation phenomenon, where the absorption satisfied the phase-matching conditions [45]. This sample exhibited the best microwave absorption property, with the maximum RL of −33.1 dB at 22.1 GHz when the thickness was 2.5 mm, and the absorption bandwidth below −10 dB is 2.2 GHz (21.3 GHz–23.5 GHz). For sample E, the maximum RL was only −14 dB at 21.6 GHz with a thickness of 0.4 mm (Figure 8c). This was mainly due to its large complex permittivity, resulting in high microwave reflection.

Figure 8 shows that when the length of GFs increased from 0.7 mm (sample D) to 1.2 mm (sample E), the microwave attenuation properties deteriorated. This can be attributed to sample E with a large permittivity in the K-band, causing an additional reflective wave on the sample surface. In other words, the microwave attenuation properties of this type of material could be controlled by the length of GFs.

## 4. Conclusions

In summary, with laser ablation, continuous and short GFs can be introduced to diamond films. By measuring the permittivity of these samples in K-band, it was found that the diamond films could be developed into a microwave attenuation material. The results showed that the sample with continuous GFs parallel to the incident electric field exhibited strong reflection characteristics, whereas the sample with continuous GFs vertical to the incident electric field exhibited poor dielectric property. From these results, it can be concluded that the samples with short GFs had better microwave absorbing property than the samples with continuous GFs. In addition, the permittivity of the latter could be modified by changing the length of GFs. Meanwhile, the diamond films after laser ablation also exhibited a high thermal conductivity.

## Figures and Tables

**Figure 1 materials-12-03700-f001:**
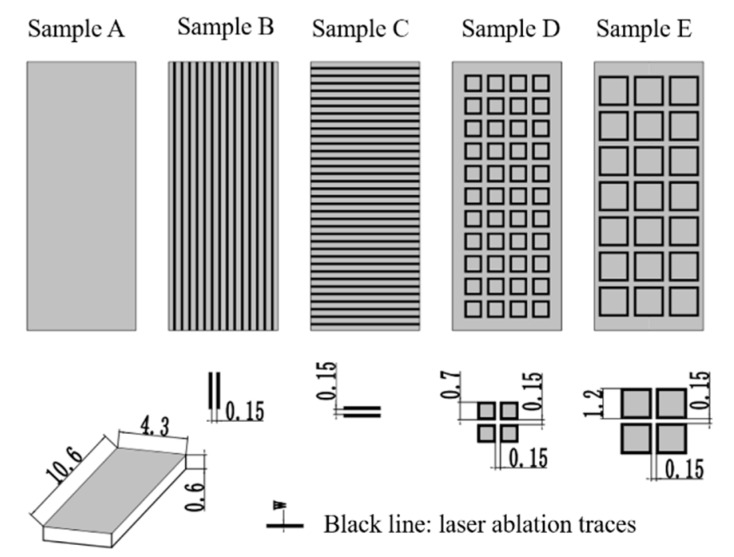
Schematic diagram of laser ablation traces. All samples have the same size of 10.6 × 4.3 × 0.6 mm^3^. Sample A was not ablated by laser. Samples B–D were ablated by laser but the ablation traces on these samples are different.

**Figure 2 materials-12-03700-f002:**
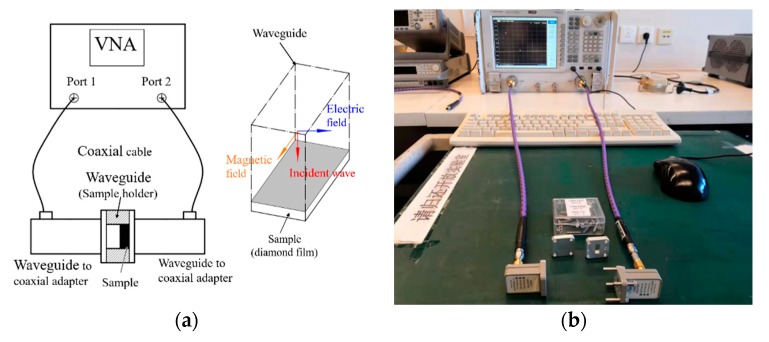
(**a**) Transmission/reflection (T/R) measurement scheme; (**b**) practical measurement system.

**Figure 3 materials-12-03700-f003:**
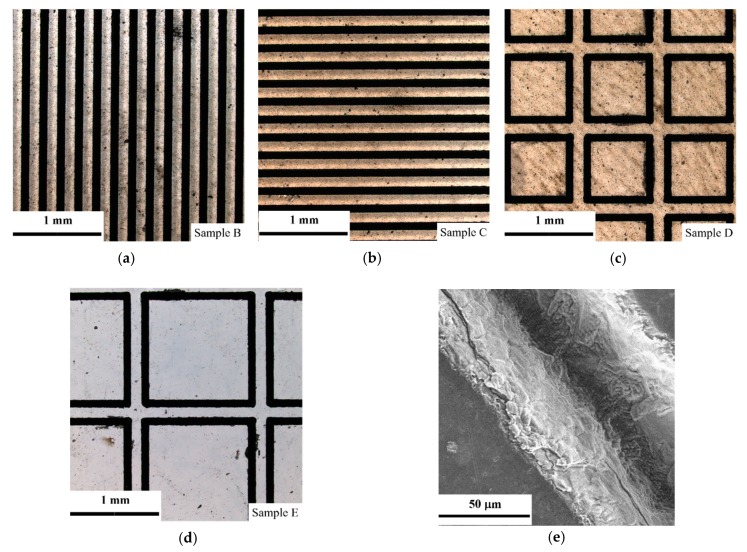
Morphologies of diamond films with graphite fibers (GFs) using an optical microscope: (**a**) Sample B; (**b**) sample C; (**c**) sample D; and (**d**) sample E. (**e**) SEM image of a laser-ablated groove.

**Figure 4 materials-12-03700-f004:**
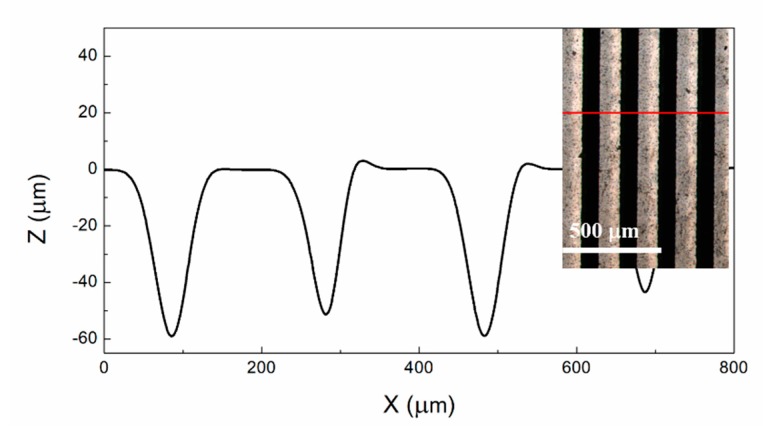
Depth of laser-ablated grooves of sample B.

**Figure 5 materials-12-03700-f005:**
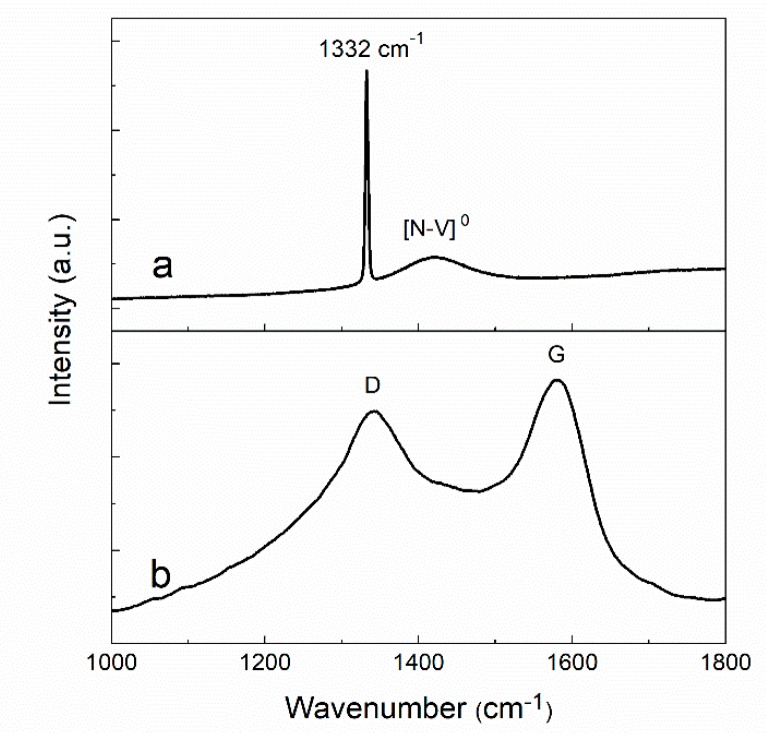
Raman spectra of (**a**) virgin surface and (**b**) inside laser-ablated area.

**Figure 6 materials-12-03700-f006:**
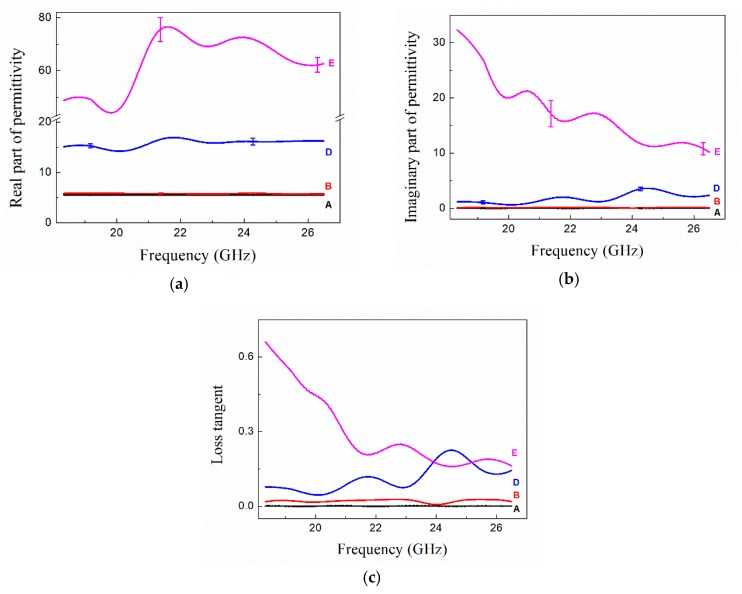
Real part (**a**) and imaginary part (**b**) of permittivity and loss tangent (**c**) of diamond films in the K-band.

**Figure 7 materials-12-03700-f007:**
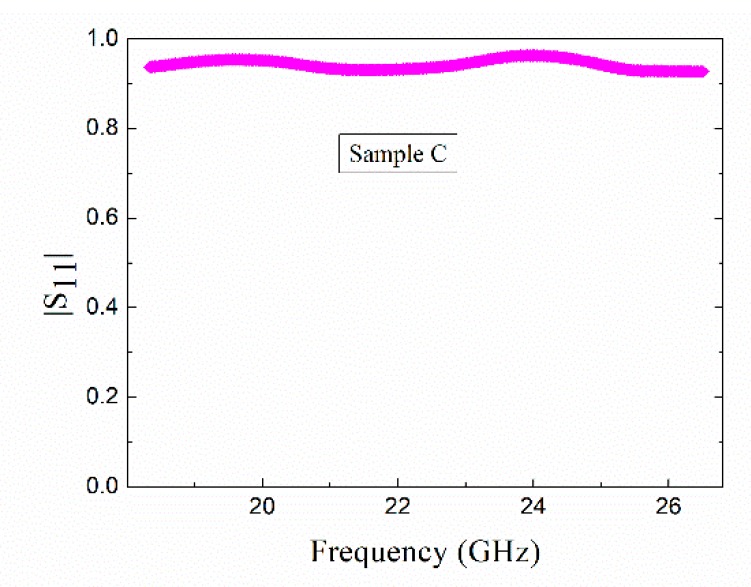
|S_11_| of sample C in the T/R measurement.

**Figure 8 materials-12-03700-f008:**
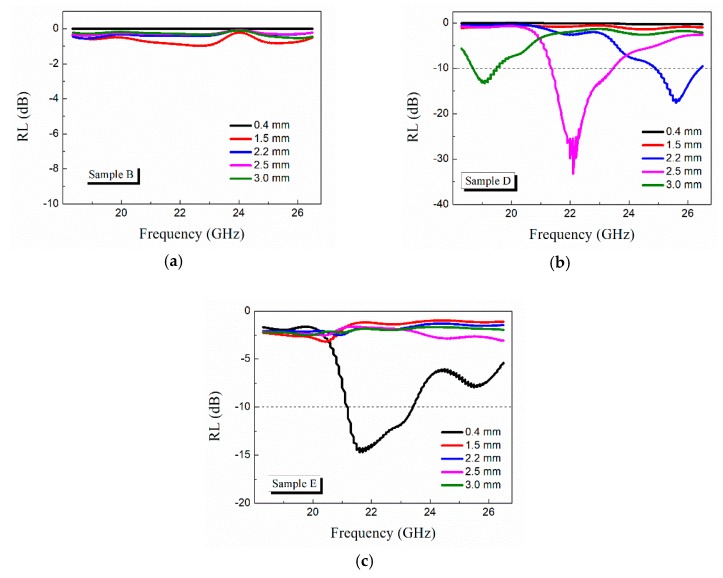
Frequency dependency of the reflection loss (RL) of samples B (**a**), D (**b**), and E (**c**) with different thickness in K-band.

**Table 1 materials-12-03700-t001:** Deposition conditions, growth rate, and dimension of diamond film sample.

Power (kW)	CH_4_/H_2_ (sccm)	Temperature (°C)	Pressure (kPa)	Thickness after Polishing (mm)	Deposition Rate (μm/h)
5.6	15/300	975	20	0.6	4.8

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
