# Peer review of "Effect of Laser Ablation on Microwave Attenuation Properties of Diamond Films"

_materials, 2019, doi:10.3390/ma12223700_

Round 1

Reviewer 1 Report

Dear authors,

While the topic of your paper and the experiments conducted appear quite interesting, there are some drawbacks.

Please include some information on:

I found the introduction to the problem quite short. What is the interest in materials with high microwave absorption? There are other interesting materials apart BeO and doped diamond? Why did you use a laser to create graphite in the diamond? I recommend extending the introduction and add more references on the topic. Why are you using that kind of laser source (500 us, 10.6 um)? Did you do any previous tests with any other lasers? The laser beam energy distribution is a gaussian one? Did you use the same process parameters in all samples? In the description of the laser system, please include the laser beam size as well as the power and process speed used.  At line 102, what do you mean with the expression ‘black materials’? The explanation about the lower thermal conductivity of your diamond samples due to N impurities maybe needs some references. The Raman results included in the paper show the presence of an amorphous carbon layer (ref 15 from the paper) but it is not clear if it is graphite or not. Please, clarify what do you mean with Graphite Fibers (GFs) and, if necessary, add some references. Why do you do not include the reflectance curves of all samples? From my point of view, the explanation is disordered. You should present first of all the R curves, describe the differences between them, and then explain that differences. I think that the explanation on lines 189-190 should be deeper, or referenced.

Overall, I think the paper presents some interesting results, but the explanation of the results is not clear. Results and conclusions are mixed and some discussion in the reflectance of the different samples will be interesting. The whole text should be redrafted.

Author Response

Thank the review for all comments.

Please see the revised manuscript and response in the documentation.

Reviewer 2 Report

The paper presents an interesting study with the graphitization of diamond films for the further use as microwaves absorbers. The results are clear. I recommend some minor changes.

1) Some small improvements to English

2) raw 29 i would avoid "urgently needed"

3) 76-81  don't know if the description of the images is relevant, i would present more reasoning behind why they choose those geometries and how can that affect the targeted applications

4) I would need more comments on the important of the laser shape geometry and how does that influence the graphitisation process. Also comment on how the impurities in the film affect the whole process. I would also like to have seen a discussion about the importance of the wavelength

Author Response

(The authors gave the same response as above.)

Reviewer 3 Report

I highly recommend to reconsider the comment and improve the text and the quality of the presentation. 

Author Response

(The authors gave the same response as above.)

Round 2

Reviewer 1 Report

The authors have answered all the previous questions and suggestions. The draft is now more clear. I just have two small suggestions:

In the calculations of the microwave absorbing property, the shown thicknesses are quite different for the different samples. How many thickness values did you use for each sample? What range of sample thickness did you use? 

Also, a final revision on the English, as sometimes the used expressions make the text a bit confusing.

Author Response

Thank the review for all comments.

Please see the revised manuscript and response in the documentation.

This manuscript is a resubmission of an earlier submission. The following is a list of the peer review reports and author responses from that submission.